# Controlling Cell Trafficking: Addressing Failures in CAR T and NK Cell Therapy of Solid Tumours

**DOI:** 10.3390/cancers14040978

**Published:** 2022-02-15

**Authors:** Lydia G. White, Hannah E. Goy, Alinor J. Rose, Alexander D. McLellan

**Affiliations:** 1Department of Microbiology and Immunology, University of Otago, Dunedin 9016, New Zealand; goyha645@student.otago.ac.nz (H.E.G.); alinor.j.rose@gmail.com (A.J.R.); 2Southern Community Laboratories, 472 George Street, Dunedin 9016, New Zealand

**Keywords:** CAR T cells, solid tumours, adoptive cell therapy, chemokines, T cells, natural killer cells, vasculature

## Abstract

**Simple Summary:**

Using immune cells to treat cancers is a promising new approach for leukemias and lymphomas. However, the use of cellular therapy in solid tumours is compromised by poor migration to the tumour, the physical barriers to infiltration, and the active suppression by the tumour. We will review both past and emerging strategies that could be used to enhance the migration and infiltration of adoptively transferred cell types, including CAR NK and T cells, in solid tumour immunotherapy.

**Abstract:**

The precision guiding of endogenous or adoptively transferred lymphocytes to the solid tumour mass is obligatory for optimal anti-tumour effects and will improve patient safety. The recognition and elimination of the tumour is best achieved when anti-tumour lymphocytes are proximal to the malignant cells. For example, the regional secretion of soluble factors, cytotoxic granules, and cell-surface molecule interactions are required for the death of tumour cells and the suppression of neovasculature formation, tumour-associated suppressor, or stromal cells. The resistance of individual tumour cell clones to cellular therapy and the hostile environment of the solid tumours is a major challenge to adoptive cell therapy. We review the strategies that could be useful to overcoming insufficient immune cell migration to the tumour cell mass. We argue that existing ‘competitive’ approaches should now be revisited as complementary approaches to improve CAR T and NK cell therapy.

## 1. Introduction

Lymphocytes are highly migratory cells with the capacity to infiltrate tissues via vascular connections and direct interstitial movement. Lymphocytes migrating to solid tumours face a number of physical and biological barriers that prevent migration into the tumour margin and core. Bypassing these physical and chemical barriers and overcoming the suppressive metabolic and biological factors are a requirement for an effective anti-solid-tumour response [1,2,3,4,5]. The correlations between the number of tumour-infiltrating lymphocytes and better clinical outcomes provide critical evidence highlighting the importance of guiding the migration of lymphocytes to solid tumours [6]. Moreover, the demonstration that ex vivo expanded tumour-infiltrating lymphocytes could be used to achieve the regression of melanoma highlights further the relevance of regional responses for effective tumour immunotherapy [7].

The advent of genetically engineered adoptive cell therapy (ACT) has changed the treatment landscape for haematological cancers and given promise to the use of such therapies against solid tumours. The ability to transfer genetic elements to T cells to facilitate migration in addition to tumour-antigen recognition is a hotly researched topic in experimental and clinical trials. However, overt strategies focused on stimulating chimeric antigen receptor (CAR) T or NK cell migration to solid tumours are lacking from current clinically approved approaches [8]. Such approaches are restricted by the cargo limits of the gene-transfer techniques within viral vectors or transposons and should not negatively impact the manufacturing process or interfere with the primary aim, which is to efficiently introduce a genetic element to enforce tumour-antigen recognition. Therefore, approaches such as the co-administration of soluble agents, such as small molecule agonists, checkpoint inhibitors, and stimulatory BiTEs, are attractive solutions to complement the activity of CAR T cells. Several of these approaches have been utilised in clinical trials (Table 1).

Conventional therapy, including chemo- and radio-therapy, has been shown to stimulate T cell migration into tumours for enhanced therapeutic effects, and radiotherapy allows abscopal effects on distant metastases [9,10,11,12]. Similarly, direct induction of inflammatory events within the tumour, using the intra-tumoural administration of pathogen-associated molecular patterns or oncolytic viruses, stimulates both systemic and local events at the tumour site to encourage lymphocyte migration and antigen-specific anti-tumour responses [12]. Moreover, antigen recognition alone is a major stimulus for migration, due to the arrest of migrating lymphocytes at the site of the antigen encounter and the sensing of soluble/tethered factors that act to stimulate CAR T cells as well as bystander cell infiltration [13,14,15]. However, the lack of homogenous antigen expression on solid tumour masses means that adoptive cell therapies may lack a critical stimulus for signposting the tumour for concerted immune cell destruction. Even when the antigen is homogenously expressed, lymphocyte tolerance/exhaustion prevents effective tumour cell recognition [13]. Conversely, high-affinity antigen interactions allow the inappropriate recognition of non-malignant tissue displaying low levels of tumour-associated antigens. Such on-target/off-tumour toxicities may be overcome by antigen-receptor affinity de-tuning [16] or the incorporation of Boolean ‘AND’ or ‘NOT’ gates to suppress lymphocyte activation within non-malignant tissues [17,18]. 

Immune tolerance or ignorance of tumour cell masses may relate to it being a property of malignant cell masses that they closely resemble undifferentiated, immune-privileged tissues, as well as ongoing Darwinian cell selection to immune-evasive phenotypes [19]. As discussed within this review, direct approaches to improve T cell migration to tumours have demonstrated promising results in animal models. However, the issues of antigen expression, physical barriers, and immune suppression also need to be overcome for migration-inducing strategies to function effectively. While emphasising the importance of lymphocyte migration to tumours, it remains unclear if the strategies that induce migration will be universally effective in the face of the hostile solid tumour microenvironment (TME). Strategies that encourage both lymphocyte migration and increased effector cell activity at tumour sites are more likely to result in effective anti-tumour responses [20]. 

As early as the 1980s, chimeric antibody constructs were developed to carry toxins, chemotherapeutic drugs, cytokines, or radionuclides and were later repurposed to facilitate lymphocyte homing to solid tumours [21]. Directly targeting T cells to the tumour with anti-GD2 mAb conjugated to IL-2 improved the response of endogenous T cells and NK cells [21,22]. Tumour-targeted lymphotoxin-alpha induced the de novo formation of lymphoid tissue with high endothelial venules, improving the anti-tumour responses, even in lymphoid-organ-depleted animals [21,22,23]. Such approaches can now be revisited as complementary approaches to improve adoptive cellular therapy. 

We will review the fundamental concepts in immune cell migration and update the reader on new strategies that have the potential to enhance the localisation of cellular therapy to the tumour. In addition to directly enhancing lymphocyte migration, we will also summarise the strategies that encourage the infiltration of antigen-presenting cells, including dendritic cells, whilst preventing suppressive immune cell infiltrations. 

## 2. Chemokines and their Receptors

Chemokines are low molecular weight cytokines that regulate immune cell migration and trafficking [24]. Chemotactic gradients are essential to recruit effector cells to sites of inflammation, including the tumour microenvironment (TME). The chemokine expression profile of solid tumours is made up of chemokines secreted by stromal cells, tumour cells, and tumour-associated immune cells and determines what immune cells are recruited to the TME, either helping or hindering tumour growth [25]. However, chemokine/chemokine receptor mismatches between the tumour chemokine profile and cognate G protein coupled receptor (GPCR) expression on the appropriate immune cells limits anti-tumoural immune cell infiltration [25,26]. 

Modulation of the complex chemotactic axes controlling cell migration is a potential avenue to enhance T cell localisation to tumour sites (Figure 1). A variety of chemokine/chemokine receptor strategies have been utilised in immunotherapeutic T cell preclinical work to facilitate CAR T cell targeting to tumour masses, including the exploiting the CXCR3, CXCR2, CCR5, CCR2, and CCR3 axes. There are currently no licensed chemokine immunotherapeutic strategies for cancer, although multiple avenues of research have promising clinical potential.

Alteration of the chemokine expression profile can be achieved by the direct intratumoural administration of purified recombinant chemokines, or by overexpressing chemokines at tumour sites using genetically modified oncolytic viruses. Increasing desired chemokine expression specifically at tumour sites establishes gradients recognised by circulating effector cells. This approach may be non-viable in cases of inaccessible or late-stage metastasised cancer, although some oncolytic viruses may be administered intravenously [27,28].

Alternatively, the native chemokine expression profile of the tumour could be exploited through the expression of the cognate chemokine receptors on adoptively transferred cells. The endogenous chemokine profile of the cancer can include chemokines that recruit pro-tumoural or regulatory immune cells to the tumour, directly influencing the cancer mass to favour tumour cell proliferation, survival, a stem-like phenotype, and metastases and can also influence the remodelling of the local tumour microenvironment [29,30,31,32,33,34,35]. These pro-tumoural chemokines typically do not recruit anti-tumour immune cells. In addition, in vitro culture and genetic alteration of the T cells during CAR T cell production may lead to a loss of the surface expression of the requisite chemokine receptors [25,36]. Strategies to enhance T cell infiltration can take advantage of the secretion of a variety of chemokines which do not typically recruit effector T cells, using gene transfer or the phenotypic modification of adoptively transferred cells [37,38,39,40]. For a comprehensive review of the role of various chemokines in tumour progression, readers are referred to Do et al. [41].

### 2.1. CXCR3 

The CXCR3 axis is critical pathway for immune cell recruitment to solid tumours and has been manipulated to increase the anti-tumour response. The expression of CXCR3 is induced and highly expressed on effector T_H_1 polarised CD4^+^ T cells, CD8^+^ cytotoxic T cell lymphocytes, NK cells, and NK-T cells following activation and binds to IFNγ-induced ligands CXCL9, CXCL10, and CXCL11 [42]. The expression of CXCR3 ligands by the tumour, the elevated serum levels of CXCR3 ligands, or CXCR3 on T cells enhances T cell recruitment and positive clinical outcomes in a range of cancers [43,44,45,46,47]. The tumour-localised expression of CXCL11 by an oncolytic vaccinia virus in a mouse model of mesothelioma successfully increased the trafficking of endogenous cytotoxic T lymphocytes to the tumour and induced systemic anti-tumour immunity, highlighting the importance of the CXCR3 axis for lymphocyte migration [48]. The expression of other CXCR3 ligands, CXCL9 and CXCL10, has been shown to be epigenetically repressed in colon and ovarian tumour cells via DNA methylation and the H3K27me3 activity of the polycomb repressive complex 2 [49,50]. The indirect and direct inhibition of PRC2 component EZH2 (DZNep and GSK126, respectively) and DNA methyltransferase 1 inhibition (5-aza-2′-deoxycytidine) made it possible to alleviate CXCR3 ligand repression and enhance CD8^+^ T cell migration and infiltration [49,50]. The use of epigenetic modulators (DNA methylation and H3K27 trimethylation inhibitors) restored the tumoural expression of CXCL9 and CXCL10, enhancing CAR T cell efficacy in a murine model of ovarian cancer, with improved T cell infiltration and tumour growth control [49]. The intratumoural injection of oncolytic adenovirus expressing CXCL10 enhanced ACT and eradicated well-established tumours in a mouse model of myeloma [51]. The co-injection of two oncolytic adenoviruses encoding CXCL10 and IL-18 into established tumours resulted in reduced tumour growth in a mouse myeloma tumour model, with complete tumour regression in 80% of the mice, although CXCL10 virus injection alone only slowed the tumour growth, without complete regression. This suggests that singularly targeting the CXCR3 axis is insufficient for treatment of all cancers [52]. A comparison of CXCL11 delivered by either intratumoural injection of CXCL11 modified vaccinia virus or by CAR T cells modified to secrete CXCL11 showed that the vaccinia virus delivery was more effective in enhancing CAR T cell treatment [53]. CXCL11 production by CAR T cells themselves likely had little effect due to the inability of CAR T cells to establish localised gradients at tumour sites [53]. The CXCR3 axis could also be indirectly targeted by the inhibition of PKA in T cells, via the expression of intracellular inhibitor regulatory subunit anchoring disruptor (RIAD)—a small peptide that inhibits the association of PKA and ezrin, resulting in increased CXCR3, increasing the tumour infiltration and treatment efficacy of CAR T cell treatment [54].

### 2.2. CXCR2

The modulation of CXCR2 expression on T cells has been shown to increase treatment efficacy in a range of tumour models due to the enhancement of T cell trafficking. CXCR2 is not normally expressed on T cells but is specific for a variety of ligands overexpressed on a range of tumours, including CXCL1, CXCL2, CXCL5, and CXCL8. These ligands are also expressed by various infiltrating immune cells and are typically considered pro-tumoural chemokines. For example, CXCL8 can promote invasion and metastasis directly and can promote a pro-metastatic niche by recruiting granulocytic myeloid-derived suppressor cells and neutrophils [25]. A strategy to exploit the expression of pro-tumoural cytokines is to ectopically express their receptors, such as CXCR2 on desired effector cells, thereby targeting them to the tumour. The expression of CXCR2 in primary T cells enhanced migration toward both recombinant and melanoma-tumour-derived CXCL1 in vitro, and the surface expression of CXCR2 enhanced the accumulation of T cells in tumours in a mouse model of melanoma, improving tumour regression and survival [55,56]. An ongoing phase I/II trial aimed at treating metastatic melanoma patients with tumour-infiltrating lymphocytes transduced with CXCR2, followed by high dose IL-2 has clinical outcome reports pending (NCT01740557). CXCR2 expression has also shown promise in application to CAR T cell treatment. Hepatocellular carcinoma tumours secrete various CXCR2-binding chemokines, but CXCR2 is not expressed on peripheral T cells or tumour infiltrating cells. The enforced expression of CXCR2 on CAR T cells increased migration and accumulation within tumours in a hepatocellular carcinoma tumour model, improving treatment outcomes [37]. Similarly the ectopic expression of CXCR1, which is not normally expressed on T cells, or CXCR2, on anti-CD70 CAR T cells, enhanced the trafficking to tumours and elicited tumour regression and improved survival in models of glioblastoma, pancreatic cancer, and ovarian cancer [38]. Mice that received CAR T cells ectopically expressing chemokine receptors after retroviral transduction with transgenes responded well to tumour rechallenge, with complete regression and surges of CXCR1^+^ and CXCR2^+^ CAR T cells at the tumour localities, indicating a long-lasting tumour control capability of the chemokine receptor expressing T cells [38]. This particular modified CAR T cell treatment synergises well with ionising radiation, which enhances tumour expression of CXCR1/2 ligand IL-8 [38]. Critically, IL-8 has been shown to have pro-tumoural effects; so, strategies that utilise IL-8, such as ectopic CXCR1/2 expression, must acknowledge the possible tumour potentiating effects [57].

### 2.3. CCR4

The receptor CCR4 has low expression on CD8^+^ T cells, but its ligands CCL22 and CCL17 recruit pro-tumoural T_H_2 and regulatory T cells to the tumour to support tumour growth. Enforcing the expression of CCR4 via retroviral transduction of anti-CD30 CAR T cells enhanced migration to lymphoma cells in vitro and increased the efficacy of CAR T cell treatment in a mouse model of Hodgkin’s Lymphoma [39]. Although this work used a haematological cancer model, this approach could also be applied to solid tumours as CCL22 and/or CCL17 expression upregulation has been identified in gastrointestinal, ovarian, and pancreatic cancers [58,59,60]. In a murine model of pancreatic cancer, the retroviral transduction of antigen-specific cytotoxic T cells with CCR4 enhanced migration to tumour sites and eliminated tumours in 40% of mice [61]. Interestingly, the authors also show that modification with CCR4 enhances interaction with DCs, with the strengthening of T cell LFA-1 binding to DC ICAM-1. This strategy therefore has the potential to enhance not only infiltration of tumours but also activation and support inside the tumour [61].

### 2.4. CCL2

CCL2 is secreted by a range of tumours and tumour-supporting immune cells and induces the migration of pro-tumoral immune cells such as macrophages, T_H_2, and regulatory T cells. However, its cognate receptor CCR2b is only weakly expressed on activated T cells. The expression of CCR2b on GDH-CAR T cells enhanced trafficking to CCL2-secreting neuroblastoma 10-fold, compared to non-CCR2b-expressing CAR T cells, and increased the relative anti-tumour activity [40]. However, targeting the CCL2 pathway may have some unintended toxicity consequences as CCL2 can be expressed by a range of non-tumour cell types, including endothelium, smooth muscle, and fibroblasts and is implicated in multiple conditions, such as rheumatoid arthritis, asthma, inflammatory bowel disease, and SARS-CoV-2 infection [62,63]. However, Craddock et al. suggest that the targeting of non-malignant tissue expressing CCL2 by CCR2b-transduced CAR T cells may not be an issue given the high level of CCL2 produced by tumours as compared to that of normal tissue [40], although this has yet to be experimentally evaluated. 

### 2.5. CXCR4

The chemokine receptor CXCR4 is widely expressed on range of haematological and solid tumours, as well as cancer stem cells. The binding of the canonical CXCR4 ligand CXCL12 (SDF1) promotes tumour cell proliferation, survival, and metastasis, with the CXCR4^+^ tumour cells having a high self-renewal capacity and potent tumorigencity [25,64,65,66]. Given the pro-tumoural role of signalling through the CXCL11–CXCR4 pathway, combining ACT with CXCR4 inhibition is a promising approach, especially given the range of CXCR4 antagonists in clinical development [67,68]. The formation of tertiary lymphoid structures within solid tumour masses can be inhibited by the action of cancer-associated fibroblast-secreted CXCL12. The inhibition of CXCR4 using continuous infusion of the small-molecule inhibitor AMD3100 (plerixafor) relieved this suppression, resulting in enhanced T and NK cell infiltration with a concomitant decrease in the recruitment of cancer-associated fibroblasts (CAF) [69]. CXCR4 inhibition is therefore a potential strategy to enhance CAR T and NK cell infiltration into tumour sites. In addition, CXCR4 has been identified as a key chemokine receptor which attracts pro-tumoural myeloid-derived suppressor cells (MDSC) to the tumour site [70,71,72]. Recent work showed enhanced efficacy using a combination of anti-EGFRvIII CAR T cells and the poly(ADP-ribose) polymerase inhibitor olaparib. Olaparib was shown to reduce CXCL12 secretion by CAF, thereby decreasing migration and infiltration of MDSC, increasing the efficacy of CAR T cell treatment in a mouse model of breast cancer [73].

### 2.6. CCL5

CCL5 is recognised by a range of chemokine receptors that are maintained on ex vivo stimulated T cells such as CCR1, CCR3, and CCR5, making CCL5 a good candidate for tumour-specific expression to target adoptively transferred lymphocyte migration [74]. CCL5 is a high-affinity ligand for CCR5, which is normally expressed on a range of immune cells, such as T, NK, innate lymphoid cells, and some subtypes of macrophage and dendritic cells [40,75]. CCL5-armed oncolytic virus and CAR T cell therapy have been successfully used in combination in a mouse model of neuroblastoma [74] (see next section).

Exploitation of the CCL5 axis has also been shown to increase efficacy in preclinical mouse models of NK immunotherapy. Infused NK cells were engineered to overexpress CCR5, and a vaccinia virus armed with CCL5 was injected intratumorally, resulting in increased NK cell infiltration and tumour regression [76]. CCL3, which also binds to CCR5, has also been investigated for targeted localisation. Intratumoural injection of adenovirus vector armed with CCL3, followed by the adoptive transfer of activated T cells, delayed tumour growth significantly, although survival was not heightened compared to adoptive transfer with a control virus in a mouse model [77].

Although the modulation of chemokine axes has shown promise in immunotherapy models, there are serious application concerns that could limit effectiveness. A critical consideration is the potential for non-specific recruitment due to the redundancy built into most chemokine axes. For example, although CCL5 can enhance the anti-tumoural immune response by the recruitment of CCR5^+^ cells such as T, NK and DC cells, it has also been shown to recruit monocytes, macrophages, and regulatory T cells and promote cancer invasion and cancer stem cells [25,78,79]. In conclusion, the roles of the chosen chemokines need to be well defined to prevent unintended recruitment of suppressive cells into the tumour. 

## 3. Oncolytic Viruses

Oncolytic viruses (OV) have the potential to be used in combinatorial strategies alongside adoptive cell therapy owing to their profound immunomodulatory effects (Figure 1). Although oncolytic viruses may infect a range of cancerous and non-cancerous cells, their replication in malignant cells is magnitudes greater than in normal healthy cells [80,81,82]. Selective replication within tumour cells is due to natural occurrence or genetic modification that enforces a greater reliance on host factors, such as free nucleosides that are more abundant or accessible in cancerous cells, or the viral exploitation of permissive cancer cells with dysregulated signalling pathways and cell cycle flux [83,84,85,86]. A range of oncolytic viruses have been and continue to be tested in clinical trial settings including: oncolytic adenoviruses, herpesvirus, reovirus, poxvirus, paramyxovirus picornaviruses, parvoviruses, and rhabdoviruses [28,87]. There are currently two approved oncolytic viral treatments, talimogene laherparepvec (T-VEC), a GM-CSF-modified recombinant herpes simplex type 1 virus which is FDA-approved for melanoma, and H101, a modified adenovirus approved in China for nasopharyngeal carcinoma alongside chemotherapy [88,89]. 

The ability of OVs to modulate the TME and influence the host anti-tumoural immune response provides strong rationale to use OVs to target CAR T cells to tumours and improve their function. Mechanistically, OVs exert their anti-tumoural function by direct lysis of cancer cells and enhancement of the anti-tumoural immune response. OVs induce the immunogenic cell death of tumour cells, with the release of damage- and pathogen-associated molecular patterns (DAMPs and PAMPs) and tumour-associated antigens or neoantigens and the induction of cytokine and chemokine expression at tumour sites [90,91,92,93,94]. OVs can induce major histocompatibility complex (MHC) expression on tumour cells, thereby increasing antigen presentation and combating immunosurveillance evasion via MHC downregulation [95,96]. OV-mediated TME remodelling influences innate and adaptive immune cell functions, enhancing dendritic cell activation and lymphocyte infiltration and activation and stimulating epitope spreading [91,97,98,99]. 

The preclinical evidence suggests that treatment with a combination of OV and ACT has enhanced efficacy compared to stand-alone treatment [100,101,102]. The intratumoural injection of a HSV-2 OV resulted in the upregulation of a range of chemokines, including crucial T cell attractant CXCR3 ligands CXCL9 and CXCL10, as well as CCL2, CCL3, and CCL4, increasing the migration of adoptively transferred T cells to tumours in a model of pancreatic adenocarcinoma [94]. In a model of melanoma, a combination of oncolytic vesicular stomatitis virus (VSV) and ACT of OT-I antigen-specific T cells exerted a synergistic effect [103]. Importantly, tumour-associated T cells had improved IFNγ production, indicating that VSV pretreatment was capable of increasing the functional capacity of T cells and controlling tumour growth [103]. A combination of intravenous oncolytic VSV vaccination and ACT using central memory polarised antigen-specific T cells in a mouse model of fibrosarcoma showed an enhanced expansion and tumour infiltration of adoptively transferred CD8^+^ T cells [100]. In addition, OV vaccination boosted the antigen-specific endogenous T cell response and supported host memory cell formation, which is crucial to prevent antigen loss and achieve durable regression [100]. An oncolytic vaccina virus expressing IL-2 enhanced the tumour infiltration of highly reactive tumour-specific endogenous CD8+ T cells in a mouse model of colon cancer, with a reduction of exhausted PD-1^hi^Tim-3^+^CD8^+^ T cells and regulatory T cells [101]. These ‘OV-induced’ tumour-infiltrating lymphocytes were expanded ex vivo and used in an ACT regime for a poorly immunogenic solid tumour and were significantly superior at controlling tumour growth [101]. The ability of OVs to attract both endogenous and adoptively transferred T cells to solid tumours and support their function and state at tumour sites demonstrates the compelling potential of using OVs and CAR T cells in combination.

Alongside the natural immunomodulation of OVs, their genetic modification allows for the tumour-localised expression of an extensive range of polypeptides or non-coding RNA to further enhance the inflammatory profile of the tumour, supporting immune cell migration and activation. A multitude of armed OVs have exhibited enhanced anti-tumour effects and are contenders for use alongside CAR T cell therapy [104,105,106,107,108]. OVs armed with an array of cytokines, chemokines, and bispecific T cell engagers (BiTEs) have been used to supplement the TME and have improved CAR T cell therapy efficacy in a range of preclinical studies [74,109,110,111,112,113]. The intratumoural delivery of the oncolytic adenovirus, armed to express chemokine CCL5 and growth factor IL-15, enhanced tumour infiltration and the persistence of anti-GD2 CAR T cells in an immunodeficient mouse model of neuroblastoma–improving tumour control and survival as compared to monotherapies [74]. Without accessory transgenes (CCL5 and IL-15), high doses of unarmed oncolytic-adenovirus controlled tumour growth without the need for CAR T cells, but synergised with CAR T cells at lower doses, with modest effects on CAR T cell persistence noted. Importantly, the expression of both CCL5 and cytokine IL-15 was required for this positive effect on CAR T cell efficacy, and simply enhancing infiltration via CCL5 expression in isolation was not sufficient to do so [74]. The tumour-localised expression of an anti-EGFR BiTE by an oncolytic adenovirus synergised with anti-FR-α CAR T cells in a mouse model of pancreatic cancer, with the enhanced accumulation of T cells at tumour sites and an increase in proliferation and activation markers expressed on the infiltrating lymphocytes [113]. An alternate use of OV alongside CAR T cell therapy is to utilise genetically modified OVs to induce target antigen expression in the tumour. This has been achieved in solid tumour immunisation regimes via the use of OVs encoding target antigens or by administering the OV alongside antigenic peptides [114,115,116]. The ability of these OVs to induce significant antigen-specific T cell responses opens the possibility of using OVs to ectopically express any antigens in tumours in the attempt to aid CAR T cell recognition, solving the problems with suitable antigen choice in solid tumours [114,115,116]. This principle was demonstrated in preclinical studies, wherein the surface CD19 antigen was delivered by OV vectors to solid tumours, resulting in efficient targeting by anti-CD19 CAR T cell therapy [117,118]. The synergy between oncolytic viruses and CAR T cell therapy has resulted in current phase I clinical trials investigating a combinatorial approach to boost the efficacy of solid cancer treatment (NCT01953900, NCT03740256).

TILT-123 is an oncolytic adenovirus armed with TNFα and IL-2 with the potential for combinatorial strategies with CAR T cells. The local expression of these cytokines promoted an anti-tumoural response by enhancing T cell proliferation, directly killing tumour cells, and by elevating the expression of other inflammatory factors [119]. The use of TILT-123 as a monotherapy and in combination with ACT has shown significant anti-tumoural activity, with modification of the tumour’s cytokine expression profile and increases in T cell infiltration, with compelling synergy with PD-L1 inhibition in a range of preclinical models [105,119,120,121,122,123]. TILT-123 is now being clinically investigated in two currently recruiting clinical trials, both as a monotherapy for solid tumours and in combination with tumour-infiltrating lymphocytes ACT for metastatic melanoma (NCT04695327, NCT04217473). Notably, this TNFα- and IL-2-armed OV demonstrated synergy with mesothelin specific CAR T cells in a preclinical mouse model of pancreatic cancer [109], supported CAR T cell activity, with enhanced tumour regression and survival and prevention of metastases. Pretreatment with this OV remodelled the tumour microenvironment, with upregulation of the markers of M1 macrophage polarisation, DC maturation, and chemokines that attract T, NK, and DCs, resulting in increased CAR T and endogenous T cell infiltration of the tumour and function [109]. This study provides strong rationale for the use of TNFα- and IL-2-armed OV in combination with CAR T cells for a range of solid tumours. 

CAdVEC is an armed OV system capable of synergising with CAR T cell therapy. This combinatorial strategy utilises both an oncolytic adenovirus and a helper-dependent adenovirus with 34 kb of immunostimulatory or immunomodulatory transgene carrying capacity [110]. CAdVEC has been utilised to locally express mini anti-PDL-1 bodies, IL-12p70, and a BiTE targeting CD44 variant 6 isoform (CD44v6) overexpressed on head and neck squamous cell carcinoma (HNSCC) [110,111,112]. These accessory transgenes overcome the limitations of CAR T cell therapy by preventing checkpoint inhibition, supporting T cell function, promoting T_H_1 polarisation, and maintaining these populations in the presence of IFNγ, while combatting tumour heterogeneity by targeting additional TAA [124]. CAdVEC expressing different combinations of these molecules at tumour sites synergised with anti-HER2 CAR T cells in models of prostate cancer and both xenograft and orthotopic (primary and metastatic) HNSCC [110,111,112]. CAdVEC harbouring accessory genes enhanced the efficacy of CAR T cell activity against solid tumours, with advanced tumour control and an increase in CAR T cell survival [110,111,112]. The addition of an anti-CD44v6 BiTE into a dual CAdVEC regime expressing both mini anti-PDL-1 bodies and IL-12p70 did not further enhance CAR T cell activity compared to the dual regime in models of pancreatic cancer and HNSCC [112]. However, the expression of all three factors in a cancer model that lacked antigen targeted by CAR T cells resulted in significant long-term tumour control and increased survival [112]. The ability for BiTEs targeting alternative antigens to allow a response against nominal antigen-deficient tumours may aid CAR T cells to combat tumour heterogeneity [112]. The intratumoural injection of CAdVEC modified to express immunostimulatory molecules will be used now in a recruiting phase 1 clinical trial alongside autologous anti-HER2 CAR T cell therapy for solid cancers (NCT03740256). The CAdVEC system could feasibly be modified to express a range of modulatory transgenes to enhance CAR T cell efficacy, target cells to the tumour, and combat immunosuppression.

OVs are a strong contender for aiding CAR T/NK cell treatment to overcome solid tumour-associated barriers, as OV may disrupt and debulk the tumour to influence the TME profile. Accessory transgene selection is critical owing to potential unintended effects, as exhibited by the expression of IFNβ by a VSV-based OV. Although this armed OV favourably remodelled the tumour chemokine profile, CAR T cell treatment efficacy was reduced, owing to type 1 IFN-induced apoptosis, inhibitory receptor expression, and dysregulation of CAR expression [125]. 

## 4. Tumour-Inducible Promotors 

Control of gene expression in ACT has been explored to further enhance the targeting of solid tumours (Figure 1). While the constitutive gene expression of additional transgenes has been used to enhance CAR T cell therapy, the specific targeting of gene expression via the use of inducible promoter systems allows for the tumour-specific expression of genes of interest. The site-specific release of inflammatory mediators, including chemokines, through tumour-antigen triggering is a useful modality to encourage the migration of other CAR T cells and bystander cells to the tumour.

Different variations of inducible systems have been tested in ACT, including T cell promoters activated by CAR-signalling, tumour metabolites, or by the clinician using drug control. Inducible systems have shown promising results in multiple preclinical studies and have recently progressed to human clinical trials [126]. The overexpression of genes such as IL-12 or IL-18 or the checkpoint inhibitors in CAR T cells has improved CAR T cell efficacy and safety [127,128,129,130,131,132]. 

Inducible promoters activated by the recognition of a tumour antigen, whether by CAR or TCR in T cells or TILs, have been used to overexpress a variety of genes, particularly anti-tumour cytokines, and have been shown to enhance the anti-tumour effect while improving safety [126,127,129,133,134,135].

Cell-activation-induced promoters make use of elements found in natural promoters, such as the IL-2 minimal promoter with the addition of transcription factor binding sites, such as the nuclear factor of activated T cells (NFAT), NFKB, or AP-1, that are activated upon T cell activation [136,137,138]. These promoters depend on the recognition of the target antigen by the CAR to trigger activation. Cell-activation-inducible promoters should, in theory, target the expression of these genes to only the tumour in a temporally and spatially confined manner. 

While IL-12 has been shown to enhance CAR T cell therapy in vivo, toxicity has been noted with systemically administered IL-12. The pre-clinical testing of targeted expression has mostly comprised IL-2-based NFAT promoters targeting IL-12 expression to the tumour site. The use of these antigen-triggered promoters to fixate IL-12 expression at the tumour site improves the CAR T cell killing of tumours in vivo and reduces the number of CAR T cells needed for effective tumour elimination. The strategy minimised the toxicity observed with systemic IL-12 expression [126,127,130,139,140]. 

NFAT-inducible promoters have shown benefit in non-CAR ACTs, such as TCR and TIL ACT, which utilise the T cells recombinant/endogenous TCR, in addition to other genes [126,133]. TILs engineered to selectively express IL-12 at tumour sites generated an enhanced anti-tumour immune response, when compared with non-engineered TILs, and required 10- to 100-fold lower doses in a clinical trial for metastatic melanoma. Although some toxicity was observed, this trial provides a promising example of inducible systems in use in humans (NCT01236573) [126]. Local expression of IL-12 by CAR T cells has also been shown to activate innate immune cells such as macrophages at the tumour site, leading to an antigen-independent innate response to tumour cells, aiding the elimination of antigen-negative tumour cells [127,129]. 

Inducible IL-18 expression by an NFAT-IL-2 minimal promoter led to an increase in effector T cells with high T-Bet and a decrease in FOX-01 T cells that were resistant to anergy induction [137]. IL-18 was shown to promote the anti-tumour activity of the CAR T cells through a widespread change to the immune profile at the TME, with a decrease in suppressive cells, including Tregs, CD103+ DCs, and M2 macrophages. This change in immune profile resulted in successful anti-tumor activity against lung and pancreatic tumours which were refractory against the same treatment without the IL-18. In another study, antigen-inducible IL-18 expression in engineered T cells led to a safe and effective anti-tumour response against a melanoma mouse model with the development of a favourable profile of T cell co-stimulatory and inhibitory receptors [129].

The development of promoters that respond to signals distinct from the TAA is another developing field of research that can take advantage of alterations in tumour metabolite signatures. He et al. have developed a hypoxic tumour microenvironment sensor using hypoxia-induced elements within a minimal CMV promoter [141,142]. The “HiTA-system” limited CAR T cell gene expression to the hypoxic environment, while the ignoring normoxic tissues that expressed the TAA. Hypoxia-inducible expression systems could also be used to express other transgenes or the CAR itself to focus and enhance the immune response [134].

While the Hi-TA system described above requires the tumour to be appropriately hypoxic, the use of clinician-controlled activator signals may also provide a way to target CAR expression to the tumour. Huang et al. have developed a light-responsive promoter trigger CAR activation of expression only in the presence tissue-penetrating blue light. Light-inducible CAR expression gave effective cytotoxicity against tumours in NSG mice [135]. Although this system is limited by how far the light can penetrate tissue, it may be used for the treatment of melanomas and other skin cancers underlying shallow surfaces [143]. 

## 5. CAR Specificity and Structural Modifications

As noted above, the inflammatory responses induced by antigen triggering are crucial for bystander CAR and non-CAR lymphocyte migration to the tumour. Therefore, the choice and design of the CAR structure is a critical consideration for optimising migration as well as anti-tumour activity. In conventional CAR T cell therapy, CAR T cells express one receptor that recognises a single tumour-associated antigen. In the case of one FDA-approved CAR T cell therapy for B cell malignancies, CD19 is targeted and allows for complete ablation of malignant and non-malignant B cells from the patient [144]. With solid tumours, it has proven to be more difficult to find a specific TAA to avoid the “on target, off tumour” destruction of non-malignant tissue by the CAR T cells. Alternatively, the antigen may be insufficiently expressed by the majority of the tumour cells in a solid cancer mass [145,146]. 

Antigen escape is a well-established concern with CAR T cell therapy, with treatment resulting in the selection of malignant cells that express low, or no, antigen. This has been observed with the downregulation of CD19 in patients with recurrent disease and in a decrease in BCMA expression in residual malignant cells in a multiple myeloma clinical trial (NCT02546167) [144,147,148]. The use of systems such as DUAL CARS and Universal CARS (UniCAR) may overcome these issues, as they can target multiple antigens at once. 

While conventional CAR T cell therapy utilizes a CAR targeting an antigen expressed on the surface of the cancer cells [149,150], the UniCAR introduces a second component known as a Targeting Module (TM). TMs are fusion molecules of the UniCAR target epitope and a tumour-antigen-binding moiety (e.g., scFv). TMs bridge the CAR to the TAA creating an “on-off switch” for clinician control of the immune response. Removal of the TM rapidly ablates cytotoxic T cell activity due to the short TM elimination half-life (15–45 min) [151,152,153,154,155].

Epidermal growth factor receptor (EGFR) is commonly overexpressed on many types of epithelial malignancies, including lung, kidney, and colon cancers; however, it is also expressed on healthy tissue, leading to an increase in the chances for “on-target, off-tumour” effects [156]. EGFR UniCARs have been successfully used both in vitro and in vivo to eliminate tumours [152,154,157]. In vivo studies have shown that the TM is rapidly cleared, allowing for an “on/off switch” [154]. 

PMSA and PSCA are expressed in prostate cancer and UniCARs targeting these solid tumour antigens have shown efficient target-specific activation of the CAR T cells, with successful tumour cell lysis in in vivo pre-clinical trials [155,158]. This in vivo work showed the safe and effective clearing of tumours when two (PCSA and PMSA) TMs were used with just one CAR demonstrating the ability of UniCARs to successfully target heterogenetic tumour cells. Radioresistant cells, such as those commonly found in HNSCC patients following radiotherapy, are often more aggressive and are associated with high fatality rates [159]. A UniCAR system targeting EGFR and CD98 through the use of two TMs was shown to significantly reduce the tumour growth of these radioresistant cells in vivo [160]. 

Several UniCAR systems have reached clinical trial, with almost all focused on haematological malignancies targeting a range of targets, including CD19 (NCT02808442 and NCT02746952), CD22 (NCT04150497), and CD123 (NCT03190278). While the CD19 trails have largely been successful, with 67% of patients achieving complete response [161], other trials, such as the CD123 UniCAR trial (NCT04106076), have experienced safety issues, including a fatality. Despite these safety issues the UniCAR system is still progressing through clinical trials. 

For solid tumours, there are only a few UniCAR clinical trials registered. One trial is targeting PMSA in prostate cancer (NCT04633148), while another ALL study (NCT03398967) utilised a UniCAR that targets either CD19/20 or CD19/22 with complete remission in >80% of refractory ALL patients after 28 days [162]. Should it prove successful, although targeting haematological malignancies, this trial will provide proof of concept for multi-antigen targeting in solid tumours. 

DualCAR T Cells express CAR receptors that target distinct antigens and can mitigate the complications of heterogeneously expressing TAAs. The initial designs for the targeting of multiple antigens involved pooling two groups of T cells transduced with different CAR constructs to enable targeting of multiple antigens [145]. Pooling two separate pools of CAR T cells has been successful in treating breast cancer through the targeting of ErbB2 and MUC1 in vivo [146]. Newer technology for DualCAR systems involves the production of CAR T cells expressing multiple CAR receptors targeting separate antigens [163,164,165,166]. These CARs can either activate or inhibit the T cells to enhance the safety and specificity of the CAR response. 

CAR T cells expressing more than one CAR on a single T cell allow for the targeting of multiple antigens using only a single transductant T cell preparation. This approach has been successfully used in a range of solid tumour cancers in vivo, suggesting promising outcomes in future clinical trials. Dual CAR targeting of neuroblastomas (GD2, B7-H3) [163] and ovarian cancer (TAG-72, CD47) [164] all showed enhanced anti-tumour responses, reduced antigen escape and improved animal survival. In vivo testing of three antigen-targeted CAR T cells against glioblastomas (HER2, IL13Ra2, EphA2) [165,166] led to successful antitumor responses against almost 100% of the tumour cells from patient-derived glioblastoma cell lines [166]. Another system for CAR T cells expressing multiple CARs is the SynNotch system, where the first CAR is expressed constitutively but upon binding to its cognate antigen it triggers expression of the second CAR. Confining the expression of the second CAR only to CAR T cells triggered by the first antigen will enhance the localisation of full CAR T cell activity to the tumour site [167].

Further developments to the DualCAR system have focused on enhancing safety using a conventional CAR that activates the T cells upon antigen recognition, with a secondary inhibitory CAR to “switch off” the T cells, should a second self-antigen be present. This inhibitory CAR may contain signaling domains from inhibitory receptors such as CTLA-4 and PD-1 to “switch off” the T cells in the presence of an antigen found on healthy tissue [18]. Currently, only dual CARs for haematological malignancies have entered clinical trials, targeting CD19/CD22 (NCT03233854). Early results show success with response in 88% of patients with ALL entering complete remission; success was lower in patients with LBCL with 62% of patients responding but only 29% entering complete remission [168]. 

The use of Dual and UniCARs has advanced our abilities to ensure proper antigen-recognition on viable tumour cells. Potential alterations of these could further involve affinity tuning and changes in CAR structure (including spacers, FcR binding ablation, and costimulatory domains) and would optimise antigen recognition and migration to and within tumours [169].

**Table 1 cancers-14-00978-t001:** Clinical trials utilising concepts with potential to enhance CAR T and NK cell infiltration of solid tumours.

System	Cancer	Characteristics	Phase	Status	Institution	NCT	Reference
Tumour inducible promoters	Metastatic melanoma	Inducible IL-12 expression in TILs. Promising anti-tumour effects, but with toxicities	I/II	Terminated	NIH Clinical Centre, National Cancer Institute (NCI)	NCT01236573	[126]
UniCAR	R/R B-ALL	Safety and feasibility of anti-CD19 UniCAR systems to induce remission in pediatric patients	I	Completed	Institut de Recherches Internationales Servier	NCT02808442	[161]
	R/R B-ALL	Safety and tolerability of increased doses of anti-CD19 UniCARs in patients, to determine MTD and recommended dose	I	Completed	Institut de Recherches Internationales Servier	NCT02746952	[161]
	B-ALL	Dose escalation and expansion of anti-CD22 UniCAR, evaluation of safety and clinical activity	I	Recruiting	Cellectis S.A.	NCT04150497	
	R/R AML	Dose escalation and expansion of anti-CD123 UniCAR, evaluation of safety and clinical activity	I	Recruiting	Cellectis S.A.	NCT03190278	
	AML	Dose escalation and expansion of anti-CD123 UniCAR, evaluation of safety and clinical activity	I	Withdrawn	Cellectis S.A.	NCT04106076	
	Prostate Cancer	Safety, side effects, therapeutic benefit of PSMA targeting UniCAR	I	Recruiting	Cellex Patient Treatment GmbH, PHARMALOG Institut für klinische Forschung GmbH	NCT04633148	
	B cell Leukemia/Lymphoma	Feasibility and safety of UniCAR treatment targeting CD19 and CD20 or CD22	I/II	Recruiting	Chinese PLA General Hospital	NCT03398967	[162]
DualCAR	R/R B Cell Malignancies	Side effects and efficacy of CD19/CD22 targeting DualCar T cells	I	Recruiting	Crystal Mackall, MD., California Institute for Regenerative Medicine	NCT03233854	[168]
	Peritoneal Carcinoma Metastatic Pleural effusion, malignant	Safety, efficacy, response rate, and duration of response in patients treated with anti-Her2 and anti-PD-L1 CAR T cells	I	Active	Sichuan University	NCT04684459	
	R/R non-CNS solid tumors	Efficacy and toxicities for CAR T cell treatment with anti-EGFR806 CAR T cells with or without addition of anti-CD19 CRA T cells	I	Recruiting	Seattle Children’s Hospital	NCT03618381	
	R/R non-CNS Solid tumors expressing B7H3	Safety, tolerability, and efficacy of CAR T cell treatment with anti-B7H3 CAR T cells with or without addition of anti-CD19 CAR T cells	I	Recruiting	Seattle Children’s Hospital	NCT04483778	
Chemokines	Metastatic melanoma, Stage III/IV Cutaneous melanoma	Feasibility and Safety of CXCR2 and NGRF transduced TILs	I/II	Active	M.D. Anderson Cancer Center, National Cancer Institute	NCT01740557	
Oncolytic Viruses	Osteosarcoma Neuroblastoma	Safety and efficacy of anti-GD2 CART cells in combination with a varicella zoster vaccine	I	Active	Baylor College of Medicine	NCT01953900	
	Her2 positive solid tumors	Safety and efficacy of anti-HER2 CAR T cells in combination with intra-tumour injection of CAdVEC an oncolytic adenovirus	I	Recruiting	Baylor College of Medicine	NCT03740256	
	Solid tumour	Safety of TILT-123; an oncolytic virus encoding TNF a	I	Recruiting	TILT Biotherapeutics Ltd. (Helsinki, Finland)	NCT04695327	
	Metastatic melanoma	Safety TIL therapy in combination with TILT-123; an oncolytic virus encoding TNF a	I	Recruiting	TILT Biotherapeutics Ltd.	NCT04217473	
Dendritic Cells	Non-Hodgkin’s lymphoma Metastatic breast cancer HNSCC	Safety and overall response rate of a combination of targeted radiation and treatment with FIt3L, Poly-ICLC, and Pembrolizumab to improve anti-tumour dendritic cell activity	I/II	Recruiting	Icahn School of Medicin at Mount Sinai, Merck Sharp & Dohme Corp. (Kenilworth, NJ, USA), Celldex Theraputics (Phillipsburg, NJ, USA)	NCT03789097	
	Low-grade B-cell Lymphoma	Response rate, safety, and tumour specific immune response after intratumoural injection of Flt3L and poly-ICLC to recruit and activate DCs to the tumour site	I/II	Recruiting	Celldex Therapeutics	NCT01976585	
	Malignant Glioma Glioblastoma Multiforme	Dose escalation of immune stimulation (FIt3L) and direct tumour killing gene transfer delivered through adenoviral vectors	I	Completed	University of Michigan Rogel Cancer Center, Phase One Foundation	NCT01811992	
	Stage IIB-IV Melanoma	Effects of anti-cancer vaccine CDX-1401 with or without addition of CDX-301for improvement in dendric cell anti-tumour activity	II	Completed	National Cancer Institute	NCT02129075	

## 6. Bispecific Antibodies 

Bispecific antibodies are an off-the-shelf therapy marrying the specificity of two or more binding domains to bridge tumour cells to anti-cancer lymphocytes. Most bispecific antibodies target a cancer-associated antigen with one domain and an immune-stimulatory molecule on T cells with the other. Bispecific antibodies and CAR T cells share the ability to activate a cytolytic immune response against tumour cells in an MHC independent manner [170,171,172]. However, bispecific antibodies can more effectively activate the bystander pool of T cells, broadening the anti-tumour response and stimulating antigenic spread [173] (Figure 1).

Bispecific antibodies have been produced in diverse formats including those with dual heavy chain/light chain combinations or with variable regions linked by short flexible peptides (BiTEs) [174]. Molecular weight and size impacts on the circulating half-life and tissue penetration, and alteration of ADCC and complement activity is possible through the choice of Fc-isotype and/or Fc-domain mutations [172,174]. This targeted design provides bispecific antibodies with several of the benefits of CAR T cell therapy, while their simplicity removes many technical difficulties associated with cellular, personalised immunotherapies, including a reduction in the cost of therapy per patient [175,176,177].

Similar to CAR T cells, the diversification of approved bispecific antibody-immunotherapies from bloodborne to solid tumors has been hindered by the tumor microenvironment (TME) of solid tumours [178]. These barriers include secreted immunosuppressive products, physical barriers that prevent immune infiltration, proteolytic enzymes, and non-malignant cellular support networks for the tumor [178,179,180]. Nevertheless, bispecific antibodies have been produced for solid cancers: Pasotuxizumab (AMG 212) is a BiTE that targets PSMA and was reported to lead to a dose-dependent reduction in plasma PSA levels. One of two short-term responders (amongst 16 total patients) underwent regression of metastases [181]. Solitomab (MT110, AMG 110) is a EpCAM/CD3 bispecific trialed in carcinoma patients. Although Solitomab was able to induce T cell migration to duodenal cancers in patients and in mouse models, its efficacy was limited by toxicity and the development of anti-drug antibodies [181]. Other experimental platforms for the development of bispecific antibodies include CD28 and 4-1BB as alternatives and in combination with CD3-based bispecifics [182]. The VelocImmune Ig-humanised mouse together with a shared light-chain pairing technique and protein A site mutations in individual heavy chains facilitate the development of bispecific antibodies to an expanding array of tumour antigens and allow the isolation of high yields of heterodimeric bi-specific antibodies [183,184]. Testing of such antibodies may be carried out in in animal models engineered to express humanised costimulatory- and/or tumour-associated molecules [183], alleviating concerns that mouse-specific antigen targets might not faithfully recapitulate those present in humans.

For the treatment of solid tumours, an ideal bispecific antibody will partition into tumour tissue and display minimum systemic activity to prevent off-target effects and other associated toxicities. Bispecific antibodies appear to exert maximum activity on effector T cells [173,177], but tumour-localised bispecific antibodies may alter the trajectory of an anticancer immune response through the activation of other T lymphocyte subsets. Choi et al. (2013) showed that CD4^+^CD25^+^FoxP3^+^ T_regs_ could kill glioblastoma cells through the granzyme-perforin pathway when stimulated with BiTE targeting EGFRvIII, revealing the potential bispecific antibodies [182].

Gardell et al. (2020) genetically modified macrophages to infiltrate tumours and express a BiTE targeting the mutated tumour antigen EGFRvIII together with anti-CD3 [181]. This technique was innovative in harnessing the macrophage’s natural ability to penetrate and reside in the hostile tumour microenvironment, aiding in the specific delivery of bispecific antibodies intratumorally. Treatment with BiTE-secreting macrophages reduced the early tumour burden in multiple models of glioblastoma and completely prevented tumour growth when combined with IL-12 [181]. 

Bispecific antibody therapy provides the potential to target more than one tumour antigen, increasing the local migration and activation of lymphocytes while decreasing tumour immune escape due to antigen mosaicism or antigen loss. Administering multiple bispecific antibody constructs or co-administration alongside other tumour-targeting therapy modalities, including CAR T cells, are additional possibilities. NanoBiTE liposomes provide recognition of CD3 and multiple cancer antigens. Although these are yet to be tested in a solid tumour pre-clinical model, they provide a promising platform to mitigate tumour immune escape by tumour antigen heterogeneity [185]. 

A novel method of localising BiTE secretion from CAR T cells was recently provided by Choi et al. (2019) [15]. A bi-cistronic construct was used to express an EGFRvIII CAR and an EGFR BiTE within a T cell. Such BiTE-CAR T cells effectively targeted and killed multiple glioma cancer cells and patient-derived xenografts. Because CARs are assumed to provide greater activity compared to BiTEs, the more specific EGFRvIII scFv was utilised for the CAR while the less specific EGFR scFv was utilised for the BiTE. Not only were the CAR T cells directed against tumour cells in a dual triggering fashion, but secreted BiTE was able to activate a bystander T cell population [15]. 

## 7. Dendritic Cell Activity

Dendritic cells (DCs) contribute to the homeostasis, activation, and migration potential of the T cell compartment. Mobilisation and activation of DCs has been shown to be a key determinant of the anti-tumoural response [12,186,187,188]. The regional activation of intra-tumoural DCs has the potential to enhance inflammation, chemotaxis, and lymphocyte infiltration [189,190]. Additionally, DCs can imprint homing phenotype/s on T cells (e.g., gut, lung, or skin), and are powerful tools to exploit to ensure the correct tissue-tropism for anti-tumour T and NK cells [191,192]. As gut-derived DC or retinol alone can induce the gut-homing α4β7 integrin on T cells [191,192], the imprinting of tissue-homing phenotypes during the early in vivo expansion phase could improve the CAR T or NK cell infiltration of tumours of the digestive tract.

DCs are crucial as they elicit T cell activation and provide homeostatic maintenance of the T cell compartment. After recruitment of conventional DC to tumour masses, TAAs or neoantigens are encountered and processed. DCs interact with tumour-localised T cells by presenting antigen and providing costimulatory signals, cytokines, and chemokines [193,194,195]. Most importantly, DCs also migrate to secondary lymphatic tissue and prime and activate naïve T cells in co-operation with lymph node resident DCs [187,188,190,196]. In the absence of antigen, DCs are important mediators of homeostatic T cell maintenance, providing tonic self-MHC and activatory signals that induce basal activation, increasing the antigen sensitivity of naïve T cells [197,198,199,200]. DC density and state in the TME is crucial to shaping the T cell response. Conventional DC1 cells, which attract and promote cytotoxic and T_H_1 cell responses, are associated with good prognosis, with enrichment of the cDC1 transcription profile being an indicator of patient survival in a range of cancers [187,190]. However, poor DC infiltration, incomplete differentiation, tolerogenic, or dysfunctional phenotypes limit the DCs’ ability to promote T cell activity [190,201,202]. 

A range of strategies to encourage endogenous DC engagement and T cell priming by proxy have been investigated, including the administration of antigens, adjuvants, or DC growth factors using a range of delivery mechanisms, and have previously been excellently reviewed [190]. Using strategies to stimulate and enhance the activity of endogenous DCs alongside adoptive T cell transfer would maximise T cell engagement and activation, encourage epitope spreading, and alter TME in a way that is more conducive to localised and systemic immune activity (Figure 1). DC antigen presentation of a range of tumour antigens to host T cells could combat the heterogenous nature of tumours and promote antigen spreading, thereby minimising antigen escape and treatment failure, which often occurs in the case of solid tumours [203,204]. Although CAR T cells do not require DC MHC-peptide presentation for activation, enhancing DC cell activation and expansion could be advantageous to CAR T cell therapy, considering the homeostatic role of DCs in tonic signalling [205].

The cytokine GM-CSF, essential for conventional DC mobilisation, attraction, and maturation, has been used extensively to enhance endogenous immune responses and alter the inflammatory milieu of the tumour. The immunotherapeutic potential of GM-CSF has been tested in a variety of modalities, including as a monotherapy, in combination with immune checkpoint blockade or as an adjuvant in GM-CSF-secreting allogenic cancer cell vaccines, with varied results [206]. There are currently two FDA-approved therapies that utilise GM-CSF. Talimogene laherparepvec (T-VEC) is an oncolytic virus for melanoma which encodes GM-CSF and sipuleucel-T, an autologous DC infusion for prostate cancer, stimulated ex vivo with a tumour-associated antigen fused to GM-CSF [207,208]. A recently reported oncolytic vaccinia virus armed with GM-CSF and a PD-L1 inhibitor resulted in enhanced DC infiltration and maturation in the tumour, which activated endogenous T cell responses against dominant and subdominant tumour neoantigens. This activity remodeled the local TME, affecting immune cell composition in both injected tumours and distant tumours [107]. 

Flt3L is a key DC growth factor for the mobilisation and expansion of conventional and plasmacytoid DCs and has been explored with the aim of enhancing DC generation and activation [209]. Multiple clinical trials using Flt3L with different combinations of a TLR3 agonist (poly-ICLC) and radiotherapy (NCT03789097, NCT01976585) or within an adenoviral expression platform (NCT01811992) are ongoing. DC-based vaccination has also utilised TLR3 agonist poly-ICLC and TLR7/8 agonist resiquimod combined with a fusion protein of a neoantigen linked to a monoclonal antibody to DC receptor DEC-205, which mediates the uptake and presentation of the antigen, designed to deliver antigen to DC cells [210]. In a clinical trial, this regime induced antigen-specific T cell responses in patients with a range of cancers, but tumour regression did not occur in most patients, highlighting the need to combine DC-based approaches with other approaches, such as checkpoint blockade or T cell-based immunotherapy [210]. A further clinical trial is also analysing the efficacy of the anti-DEC-205-neoantigen fusion vaccine in combination with the DC-targeted adjuvants poly-ICLC and Flt3L, with the results upcoming (NCT02129075). Engineering of CAR T cells to secrete Flt3L increased DC precursors in the bone marrow and tumour, increased intratumoural DC secreting IL-12 and TNF, and inhibited tumour growth [211]. Crucially, Flt3L secretion by CAR T cells promoted endogenous CD8^+^ T cell tumour infiltration and resulted in epitope spreading, highlighting the promise of using Flt3L to engage host DCs and enhance lymphocyte migration into solid tumours [211].

DC migration to and between the tumour and lymphatic system is controlled by a range of chemokine receptors, including CCR5, CCR6, and CCR7 [212,213]. DC migration to the tumour driven by the chemokine receptor CCR7 could be improved by use of armoured CAR T cells. Armoured CAR T cells expressing CCL19, a ligand for CCR7 which attracts both T and DC cells, alongside IL-7, was able to increase DC and T cell infiltration and colocalisation in tumours, achieving complete regression in some murine models and significant growth repression in others [214]. Strategies that target CCR5, as previously mentioned in Section 2, would also likely impact DC trafficking. Use of further ligands for chemokine receptors expressed on DCs should be explored to target DCs to solid tumours, although DC migration has not been studied as comprehensively as T cell trafficking. 

## 8. Targeting the Tumour Microenvironment 

As noted earlier in this review, targeting cancer-associated stromal cells through CAR T cells, bispecific antibodies, or chemokine inhibition may be particularly advantageous to ACT. In particular, cancer-associated fibroblasts (CAF) are a phenotypically heterogenous cell population that construct and remodel the TME extracellular matrix (ECM). CAF support tumour initiation, growth, and dissemination, secreting ECM proteins and a variety of regulatory factors that impact tumorigenesis and angiogenesis and metastases. The role of CAFs in the TME has previously been excellently reviewed [215,216]. Depleting or altering the functioning of CAFs is currently an area of intense research and has the potential to be an attractive option to enhance ACT and immunotherapies. Because CAF contribute to the ECM/cellular barrier, reducing their number and activity has the potential to synergise with CAR T/NK cell therapy to increase lymphocytes migration into the tumour mass. Moreover, several CARs have been developed that exert their anti-tumour effects by directly depleting CAF by targeting fibroblast activation protein α (FAP-α). Anti-FAP-α CAR T cell treatment has shown compelling efficacy in multiple preclinical models of a variety of solid tumours [217,218,219], although work by Tran et al., 2013 showed limited anti-tumoural effects of anti-FAP-α CAR T cell and lethal ‘on target, off-tumour’ toxicity [220]. There are two in-patient clinical trials utilising anti-FAP-α CAR T cells, a completed phase 1 trial for pleural mesothelioma (NCT01722149) and a phase 1 trial utilising a fourth generation CAR construct, armed to release IL-7 and CCL19 or IL-12 for a range of malignant tumours (NCT03932565).

Increased T cell proliferation and persistence at MDSC-enriched tumour sites was achieved in a study that utilised CAR T cells expressing both a tumour antigen-specific CAR and a chimeric costimulatory receptor targeting tumour necrosis factor-related apoptosis-inducing ligand receptor 2 (TR2) which is expressed on MDSCs [221]. A different strategy is to reduce the MDSC population by using modified NK cells targeting overexpression of the NKG2D ligands on MDSCs. Depleting tumour-localised MDSC increased efficacy and tumour infiltration of CAR T cells in a mouse model of neuroblastoma [222]. Increased infiltration of endogenous tumour-specific CD8^+^ T cells was noted by depleting suppressive tumour-associated macrophages (TAM) using anti-folate receptor β-CAR T cells [223]. Preconditioning with the TAM-targeted CAR T cells enhanced the efficacy of tumour-specific anti-mesothelin CAR T cells, with tumour regression and extended survival [223].

An RGD domain-based mediated targeting strategy to diminish tumour vasculature was investigated by Fu et al., with targeting of αvβ3 integrins that are expressed on endothelial cells of tumour neovasculature by CAR T cells [224]. The CAR was comprised of a modified ectodomain of the viper (*Echis carinatus*)-venom-derived echistatin polypeptide that has a high affinity for αvβ3 integrin on tumour-associated endothelial cells. These αvβ3 integrin-targeted CAR T cells destroyed tumour-associated blood vessels, leading to the destruction of the tumour mass in a mouse model of melanoma, but with possible adverse effects of tumour-site bleeding [224]. 

Solid tumours often display perturbed metabolism, switching from oxidative phosphorylation to aerobic glycolysis in a process termed the Warburg effect. This involves enhanced glucose uptake and aerobic glycolysis, leading to the release of immunosuppressive metabolites, including lactate acid, adenosine, and reactive oxygen species (ROS) that potently inhibit lymphocyte migration [225,226]. Even simple mechanisms of TME inhibition, such as bicarbonate administration to alleviate the acidic TME, were suggested to be of benefit in increasing the T cell infiltration of certain solid tumours during ACT [227]. However, more targeted approaches that alter central, intracellular metabolic pathways in tumour cells are more likely to exhibit wider applicability to a CAR-based therapy of solid tumours. A striking illustration of the potential of combining ACT with tumour- metabolism inhibitors is given by the clinical study of Cascone et al. [226]. The authors showed that melanoma patients that failed to respond to ACT exhibited upregulated tumour glycolytic activity and that inhibition of the glycolytic pathways increased the T cell killing of tumour cells. A limitation of this approach is the possible undesirable effects on the metabolism on effector CAR T cells that might rely on oxidative glycolysis [228]. For a comprehensive review of inhibitors that impact on tumour metabolism, the reader is referred to Stine et al. [229]. Overall, strategies to disperse physical and cellular barriers, inhibit tumour metabolism, and alleviate metabolite secretion have the potential to enhance the CAR T and NK cell infiltration of solid tumours. Such strategies could be targeted to alter functions in the tumour cells themselves or their associated pro-tumoural MDSC, CAF, or TAM. 

## 9. Conclusions

Targeted approaches to improving the endogenous immune responses to tumours are now more than 40 years old. The competing therapies developed in recent decades must now be explored as combination therapies to improve the disappointing performance of CAR T and NK cells against solid tumours. The option to combine agents that stimulate migration, inflammation, and tumour barrier dissolution with CAR-based therapy will provide critical avenues for advancing engineered T and NK cell therapy against solid tumours.

## Figures and Tables

**Figure 1 cancers-14-00978-f001:**
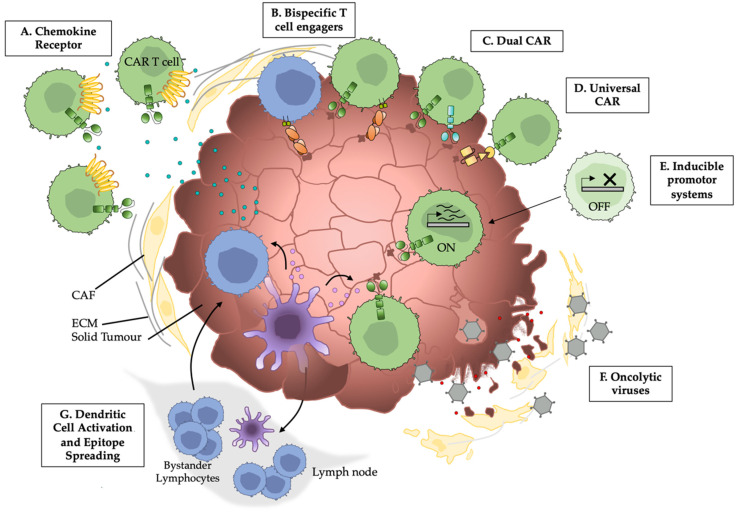
Summary of strategies to combat solid tumour physical barriers to improve CAR T/NK cell trafficking and activation. A–G represent examples of approaches to enhance the CAR T/NK cell targeting of solid tumours reviewed here.

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
