# Peer review of "Controlling Cell Trafficking: Addressing Failures in CAR T and NK Cell Therapy of Solid Tumours"

_cancers, 2022, doi:10.3390/cancers14040978_

Round 1
Reviewer 1 Report
The paper is well written. The authors made a good job to introduce new concepts in a way that can be followed by a not that much specialized audience. In the title of the manuscript as well as in the simple summary, the abstract, the introduction and the conclusions of the paper, the authors state that they'll be reviewing "strategies that could be used to enhance cellular migration and infiltration of CAR T cells in solid tumor immunotherapy". However, a small % of the strategies discussed in this paper talk actually about that. Moreover, relevant topics to be discussed under this goal are left out of the manuscript, including strategies that use adhesion proteins, strategies to target the stromal cells that are a physical barrier against the immune cells, strategies related to vascularization, or metabolism in the tumor.
I would suggest eliminating some irrelevant sections to the scope of this paper (for instance "tumor inducible promoters" are only relevant once the adoptive transferred cells are in the tumor already) or redefine the goals of the paper. However, due to the diversity of approaches discussed in this manuscript it's difficult to find a goal that encompasses them all. The authors talk about CAR T cells, TILs, different immune cells adoptive transferred, strategies to improve the endogenous immune system responses, etc...
Sections 4 (Chemokines and their receptors), 5 (oncolytic viruses) and 6 (Bispecific antibodies) are the sections that are more aligned with the scope described in the abstract the present manuscript. However, the way the topics are broken down into sections creates some overlapping between them. For instance, most of the chemokine strategies are achieved by using oncolytic viruses.
Abstract:
Line 17-19: It seems there's a word missing. Did you mean "tumor-associated suppressor macrophages"?
Graphical abstract:
Keeping in mind a maybe not that much specialized audience, the graphical abstract would benefit from a key indicating which kind of cells are depicted (anti-tumor lymphocytes, vasculature, normal tissue cells, fibroblasts, tumor cells, dying tumor cells)
Intro: the way table 1 is introduced it seems that the table is only going to list clinical trials that use CAR Ts in combination with other therapies to enhance CAR T cell migration to solid tumors. But then some of the clinical trials don't use ACT like ref. 66. Also there's some cases in which the agent combination with the adoptive cell transferred cells is not meant to increase the migration of the cells, but maybe the efficacy once in the tumor like ref. 23.
Section 2 Tumor-inducible promoters: the authors made a good job reviewing some promising strategies using inducible promoters. However, this approach to enhance CAR Ts, TCR Ts or TILs activity is not related to their ability to traffic into the tumor. So, this whole tumor-inducible promoter section is out of the scope of the paper. I would suggest either eliminate the section all together, or reformulate the goals for the manuscript.
Section 3 CAR Specificity: Although the authors make a good review about UNI and DUAL CARs. Again, all the information in this section is irrelevant to the aim of the paper. If the goals of the manuscript don't change, this section could be improved by talking about how different CAR constructs have different effects in migration of CAR T cells into solid tumors. How changes in scFv affinity, spacers, stimulatory domains, etc. have an effect on T cell trafficking.
Section 4 Chemokines and their receptors: The authors did a great job describing the importance of chemokines and their receptors in migration and engagement of immune cells in tumor clearance which is helpful to understand the relevance of modifying those axes to improve ACT.
Line 265: reference missing.
Discussion of references 89 and 90. It's unclear the relevance of the approach to induce the migration of the "effector cells". Which kind of cells are referring to?
Line 302: please clarify what the authors refer to by "positive effector cells"
Line 303-305: unclear what it means, unclear what it refers to.
Line 316: in this study only the armed vaccinia virus is used, not in combination with ACT as in the rest of the studies in that paragraph. Maybe this discussion can be moved to the beginning of 4.1. section to illustrate the relevance of the CXCR3 axis.
Line 320: wrong reference. Did you mean to cite this one? Liu, Y., Huang, H., Saxena, A. et al. Intratumoral coinjection of two adenoviral vectors expressing functional interleukin-18 and inducible protein-10, respectively, synergizes to facilitate regression of established tumors. Cancer Gene Ther 9, 533–542 (2002). https://doi.org/10.1038/sj.cgt.7700466
Is the point made in line 384 relevant when expressing CCR2b on CAR T cells? Are those risks a concern?
What about strategies inhibiting CXCR4? Is there any study worth mentioning?
Section 5 oncolytic viruses:
Line 458: Discussion of reference 128 belongs to section 4.
Line 469: none of the references cited there talk about CAR T cell therapy.
Please expand the discussion on studies ref 107, 134, 135 and 147.
Lines 503-528: the approaches discussed are about a BiTE, so this whole section could be moved to the bispecifics section.
Section 6 Bispecific antibodies: well written section. It provides a helpful revision about bispecific antibody approaches and the combination approaches.
Section 7 Dendritic cell migration and activity: again, only one of the strategies discussed in this section is about improving ACT but no mention was made about this improvement was due to better trafficking of the CAR Ts.
Reviewer 2 Report
The manuscript by White et al "Controlling cell trafficking: addressing failures in CAR T cell therapy of solid tumours" provides a brief overview of several aspects of immunotherapy for solid tumors. Overall, it is well written and logically presented. There are some thought provoking aspects in some sections. Although the manuscript is well prepared, there are some issues:
1) The major concern of the manuscript is the brevity and briefness of some sections. For example, the "Chemokines and their receptors" section is interesting, but in 2022 there are already full reviews published devoted to chemokines and cancer, and specifically the CCL5/CCR5 axis (which has nearly 200 references for this single subject). The same can be said for each section, where full reviews are available. In addition, there are recent reviews that are similar to this review, for example, Zhang et al., 2022, Pharmacological research "Improving the ability of CAR-T cells to hit solid tumors: challenges and strategies". Therefore, the overall novelty for this review is lacking.
2) As stated by the authors, the goal of the manuscript is to " ..update the reader on new strategies that may be applied to enhance localisation of cellular therapy to the tumour", lines 92-94. I'm not sure oncolytic viruses and bites would be considered cellular therapy. Also, as the title is "addressing failures of CAR T cell therapy of solid tumours" there is a lack of CAR T specific issues in these sections. There is, for example, the concept of using bites and CAR T therapy, but the discussion is somewhat superficial.
3) there are numerous references to the use of CAR T cells in hematologic cancers, so a more defined focus on solid tumors would be useful. I certainly understand that proof of concept using hematologic cancers is reasonable, but there needs to be a greater effort in how these studies directly relate to solid tumors. In some/most cases it is not obvious.
4) Several sections do not provide quantitative arguments regarding the usefulness of the discussed strategies. The review is very qualitative.
5) The section on CAR Specificity and the use of Targeting Modules is very confusing. There is little discussion as to what TMs were used and what the TMs are.
6) The references need to be checked, for example, line 227 has reference 41 as a breast cancer study, but I think its a multiple myeloma report. If so, how many others are incorrect references?
7) The manuscript provides some issues with the current use of immunotherapy for solid tumors, but doesn't really provide a way forward.
8) There are several minor grammatical/spelling errors. None are overly concerning but combined it is an issue. A thorough reading would be suggested.
Reviewer 3 Report
Cancers-154665
The review is well structured and enriched with informative content. Some minor issues should be addressed as below:
The graphical abstract should have figure legend explaining what the drawings represent.
For section 2, it would be helpful to include a table or figure summarizing the inducible promoters.
Similar to the discussion on dendritic cells, the cancer-associated fibroblasts and tumor-associated macrophages as well as their role in immunosuppressive microenvironment should be part of the discussion.
Round 2
Reviewer 1 Report
I'd like to thank the authors for thoroughly addressing all the comments. The quality of the paper increased significantly.
Moreover, based on the rebuttal letter, now it's clear the scope and the aim for this paper. That's why the title of the paper, the summary, the abstract and the conclusions should be changed to reflect that. The way is stated right now is misleading. It sets expectations for the paper that are not fulfilled. A better approach would be to state that potential strategies to improve ACT (not only CAR T cells) are going to be discussed or brainstormed. As it was done to better define table 1.
Keeping Chemokine section and OV section separately is a good idea, those sections can be better defined to avoid overlapping that it's still happening. In the chemokine section two distinct strategies are discussed and confusingly mixed. On the one hand, there's the strategy of conferring the T cell with a chemokine receptor that matches the ligands expressed by the tumor. On the other hand, there's the strategy of modifying the tumor (with OV) so it expresses the chemokines that would attract the T cells. I suggest, keeping the T cell modification in the chemokine section and move the tumor modification strategies to the OV section. Or as I suggested in the first round of revision, merging the OV section into the corresponding sections depending on the functionality of the OV.
The authors did a good job summarizing some of the strategies related to the TME. However a final conclusion paragraph on that section is lacking.
Author Response
Please see attached comments for all reviewers

Reviewer 2 Report
The edits have enhanced the manuscript. It still lacks novelty, as there are similar reviews already published.
Author Response

(The authors gave the same response as above.)

Reviewer 3 Report
the manuscript can be accepted in present form.
Author Response

(The authors gave the same response as above.)

Round 3
Reviewer 1 Report
-